Relationship between heart rate variability parameters and inflammatory activity in patients with Crohn’s disease: a retrospective study

Wang Beibei 1 wangbeibei@gdph.org.cn
Liao Shanying 1
Huang Linlin 1
Zheng Zhongwen 1
Cui Can 2 cuican@gdph.org.cn
1 Department of Gastroenterology, Guangdong Provincial People’s Hospital (Guangdong Academy of Medical Sciences), Southern Medical University , Guangzhou , China
2 Department of Anesthesiology, Guangdong Provincial People’s Hospital (Guangdong Academy of Medical Sciences), Southern Medical University , Guangzhou , China
Guan Fanglin
Electronic publication date: 2025 Aug 18
Publication date: 2025
Volume: 13
Electronic Location ID: e19893
Received 2025 May 29; Accepted 2025 Jul 21
Copyright: © 2025 Wang et al.
Copyright year: 2025
Copyright holder: Wang et al.
License: This is an open access article distributed under the terms of the Creative Commons Attribution License, which permits unrestricted use, distribution, reproduction and adaptation in any medium and for any purpose provided that it is properly attributed. For attribution, the original author(s), title, publication source (PeerJ) and either DOI or URL of the article must be cited.
License URL: https://creativecommons.org/licenses/by/4.0/

Keywords: Crohn’s disease, Heart rate variability, Crohn’s disease activity index, Autonomic nervous system

Funding: The authors received no funding for this work.

==============================
Background

It is widely recognized that individuals with Crohn’s disease (CD) often experience impaired autonomic nervous system (ANS) function. However, the specific relationship between autonomic activity and inflammatory processes in CD remains uncertain. The objective of this study was to investigate the association between autonomic nervous function, as assessed through heart rate variability (HRV) parameters, and the Crohn’s disease activity index (CDAI).

Methods

This cross-sectional study involved 82 CD patients. Demographic and medical history data were gathered, and disease activity was evaluated using the CDAI. Short-term HRV measurements were obtained from all participants.

Results

Various adjustments to the model and trend analysis indicate that the HRV parameters are significantly negatively correlated with CDAI. After controlling for potential confounding factors, the smooth curve fitting shows that high-frequency (HF) power is negatively linearly correlated with CDAI. As the HF value increases, CDAI shows a decreasing trend. In contrast, total power (TP), low-frequency power (LF), and standard deviation of the R-R interval (SDNN) exhibit a nonlinear relationship with CDAI. To the left of the inflection point, as the values of TP, LF, and SDNN increase, CDAI shows a decreasing trend. However, to the right of the inflection point, there is no relationship between TP, LF, SDNN values, and CDAI.

Conclusion

In patients with CD, autonomic dysfunction correlates with active inflammation. Among the HRV parameters, HF, which reflects vagal-mediated activity, displayed the strongest and most consistent negative correlation with inflammatory activity.

Introduction

Crohn’s disease (CD) is a chronic, transmural inflammation involving the entire digestive tract. The cause of CD includes many factors, such as genetic or environmental factors, intestinal ecological imbalance, and mucosal immune response, but its specific pathogenic mechanism is unclear (Dolinger, Torres & Vermeire, 2024).

It is believed that the autonomic nervous system (ANS) plays an important role in the regulation of inflammatory reflex (Tracey, 2002). Heart rate variability (HRV) serves as a tool for assessing autonomic nervous system (ANS) activity. HRV parameters are categorized into those that predominantly represent parasympathetic nervous system (PNS) or vagal-mediated activity (referred to as vagal nerve-mediated HRV parameters) and those that reflect the combined influence of both the sympathetic nervous system (SNS) and the PNS (Berntson et al., 1997; Kamath & Fallen, 1993). The theory of the cholinergic anti-inflammatory pathway of the vagus nerve has led to the belief that higher HRV parameter values, especially those mediated by the vagus nerve, are related to lower levels of inflammation (Young et al., 2014; Pellissier et al., 2014; Schäfer et al., 2015).

Early studies have generally indicated that CD patients have decreased autonomic nervous function (Sharma et al., 2009; Kim, Yao & Ju, 2020). Aghdasi-Bornaun et al. (2018) reported the vagal activity was inhibited, while sympathetic nerves were dominant in CD. Engel, Ben-Horin & Beer-Gabel (2015) found that the low frequency power (LF), which mainly represents the HRV index of sympathetic activity, was negatively correlated with serum C-reactive protein (CRP), a marker associated with inflammatory activity in CD patients. Sahn et al. (2023) observed that, compared with healthy peers of the same age, baseline values of HRV parameters representing vagal tone in children with inflammatory bowel disease (IBD) were lower. Yerushalmy-Feler et al. (2022) found that in children with IBD who were in remission, the square root of the mean squared differences of successive R-R intervals (RMSSD) was notably higher compared to children experiencing active disease. Hirten et al. (2025) collected HRV parameters of CD patients via wearable devices and found no difference in the circadian pattern of SDNN during symptomatic flares vs. remission. The connection between HRV parameters and inflammatory activity in CD remains debated. Consequently, this study aimed to assess the association between ANS functioning and CD activity, as reflected by HRV parameters. If the vagus nerve is indeed involved in CD pathogenesis, HRV parameters mediated by the vagus nerve should reflect this activity.

Materials and Methods

Study design and study population

This was a retrospective study, and was approved by Research Ethics Committee of Guangdong Provincial People’s Hospital, Guangdong Academy of Medical Sciences (NO.GDREC2015048H, Date: Jan 22, 2015). All patients with CD treated at our hospital from February 1, 2015 to February 1, 2020 were included. Patient demographic information, personal history, medical history including age, scope of CD involvement, disease behavior, course of disease, history of surgery, history of drug treatment, and the results of laboratory tests was collected and reviewed. The diagnosis of Crohn’s disease is based on the standards recommended by the World Health Organization (WHO). The extent of CD was classified using the Montreal classification. The Crohn’s disease activity index (CDAI) was calculated to evaluate the degree of active inflammation.

Written informed consent was obtained from all participants. The following exclusion criteria were applied: (1) Absence of medical history and/or examination records; (2) Diagnosis of heart conditions such as congenital heart defects, valvular dysfunction, conduction abnormalities, ischemic heart disease, or any form of heart failure; (3) Presence of endocrine disorders, including thyroid dysfunction, diabetes, neuropathies, or other systemic inflammatory conditions; (4) Use of medications that influence the cardiovascular system and nerve conduction, such as antidepressants, analgesics, or glucocorticoids; (5) A history of smoking or alcohol consumption; (6) Pregnancy.

Measurement of HRV

HRV measurements were performed by the same well-trained doctor. The examination was carried out in a fixed temperature and light controlled environment to avoid interference, and all examinations were performed within a same time period (10:00 am to 1:00 pm). Participants were instructed to avoid caffeine, strong tea, nicotine or other stimulants for 48 h before the examination, and to have adequate sleep the night before the examination. Patients were also instructed to fast for at least 2 h before the examination, and rest for at least 15 min before the examination.

A short-range HRV analyzer (DLP6000; General Meditech Inc., Shenzhen, Guangdong, China) was used to record and evaluate patient sitting ECG signals, and an electrocardiogram was continuously recorded for 5 min using three leads of the standard limb lead II. During the examination, patients were required to be completely relaxed, breathe regularly, and to not talk or move. Technicians carefully observed the patients to ensure there was no subjective discomfort or motion artifacts during the recording. The device’s software (version 1.0.5649.19505) processed the data, and heart rate variability (HRV) was evaluated using both time-domain and frequency-domain approaches. Autonomic nervous disorders (vagus nerve or sympathetic nerve) related to specific diseases can be reflected in HRV indexes. Time-domain and frequency-domain HRV parameters calculated in this study are shown in Table 1.

Table 1 Definition for the time domain and frequency domain indices of heart rate variability calculated in this study.

Methods	Description	
Time domain indices		
Standard deviation of the R–R intervals (SDNN ms)	Reflects the influence of parasympathetic and sympathetic system on heart-rate variability	
Frequency domain indices		
Total power (TP ms2) (0–0.4 Hz)	Reflects total variance of R–R interval over the 5 min segment	
Low frequency power (LF ms2) (0.04–0.15 Hz)	Reflects both sympathetic and parasympathetic activity	
High frequency power (HF ms2) (0.15–0.4 Hz)	Reflects the centrally mediated parasympathetic activity	

Statistical analysis

Continuous variables that followed a normal distribution were presented as means ± standard deviations (SD). For those that were not normally distributed, data were expressed as medians with interquartile ranges, while categorical variables were displayed as frequencies and percentages. To compare differences between groups, one-way ANOVA was employed for normally distributed variables, the Kruskal-Wallis test for non-normally distributed variables, and the chi-square test for categorical variables. Univariate linear regression analysis was conducted to examine the relationships between HRV parameters and the CDAI. In accordance with the STROBE guidelines, we reported results from unadjusted, minimally adjusted, and fully adjusted models. Covariates, once incorporated into the model, resulted in at least a 10% change in the matched odds ratio and were thus adjusted (Kernan et al., 2000). Furthermore, the generalized additive model (GAM) was applied to capture potential non-linear relationships. If a non-linear correlation was identified, a two-piecewise linear regression model was utilized to assess the threshold effects of HRV parameters on CDAI, following a smooth curve. When a clear relationship between HRV parameters and CDAI appeared in the smooth curve, a recursive method was employed to determine the inflection point, optimizing the model’s likelihood (Liu et al., 2013). All statistical analyses were performed using R software packages (http://www.r-project.org, R Foundation) and EmpowerStats (http://www.empowerstats.com, X&Y Solutions, Inc., Boston, MA). A P-value of less than 0.05 (two-sided) was considered indicative of statistical significance.

Results

Patients

Of the 92 patients with CD (68 males, 24 females) identified in the medical records, 10 were excluded: three (two males, one female) with incomplete medical histories, two males with arrhythmia, four (three males, one female) who had received glucocorticoids, antidepressants, and analgesics, and one male smoker. Consequently, 82 participants (60 males, 22 females) were enrolled in the study. (Fig. 1).

Figure 1 Study participants.

Patient characteristics

Patient characteristics are summarized in Table 2. The mean age of the patients was 28.84 ± 11.87 years old, and approximately 73.2% of them were male. The patients were divided into two groups based on CDAI: active CD, CDAI > 150 (mean 248.81 ± 83.27) and remission CD, CDAI ≤ 150 (mean 94.43 ± 33.24). No substantial variations were observed between the two groups in terms of age, gender, marital status, educational attainment, Montreal classification (including age at diagnosis, disease site, and disease behavior), disease duration, history of bowel resection, albumin levels, or white blood cell (WBC) count. However, a significantly greater number of patients in the remission group received biologics for treatment. Compared with remission group, the body mass index (BMI) and HRV parameters (including TP, LF, HF, SDNN) in active group were significantly lower, while heart rate (HR) and CRP level were significantly higher.

Table 2 Demographic characteristics for study participants.

Groups	CDAI > 150 (n = 51)	CDAI ≤ 150 (n = 31)	P-value	
Age (years)	28.96 ± 11.80	28.65 ± 12.18	0.908	
Gender (n, %)			0.498	
Males	36 (70.59%)	24 (77.42%)		
Females	15 (29.41%)	7 (22.58%)		
BMI (kg/m2)	17.97 ± 2.94	20.33 ± 3.55	0.002	
Marital status (n, %)			0.986	
Unmarried	33 (64.71%)	20 (64.52%)		
Married	18 (35.29%)	11 (35.48%)		
Level of education (n, %)			0.658	
Under middle school education	12 (23.53%)	8 (25.81%)		
Senior high school education	11 (21.57%)	9 (29.03%)		
University degree or equivalent	28 (54.90%)	14 (45.16%)		
Duration of disease (years)	2.00 (1.00–6.00)	2.00 (1.55–5.00)	0.388	
History of bowel resection (n,%)			0.877	
No	37 (72.55%)	22 (70.97%)		
Yes	14 (27.45%)	9 (29.03%)		
Medications (n, %)			<0.001	
Mesalamine	37 (72.55%)	8 (25.81%)		
Biologics	12 (23.53%)	20 (64.52%)		
Immunomodulators	2 (3.92%)	3 (9.68%)		
CDAI score	248.81 ± 83.27	94.43 ± 33.24	<0.001	
Age at diagnosis (n, %)			1.000	
A1	9 (17.65%)	5 (16.13%)		
A2	36 (70.59%)	23 (74.19%)		
A3	6 (11.76%)	3 (9.68%)		
Disease location (n, %)			0.745	
L1	12 (23.53%)	7 (22.58%)		
L2	4 (7.84%)	3 (9.68%)		
L3	34 (66.67%)	19 (61.29%)		
L4	1 (1.96%)	2 (6.45%)		
Disease behavior (n,%)			0.490	
B1	31 (60.78%)	22 (70.97%)		
B2	8 (15.69%)	5 (16.13%)		
B3	12 (23.53%)	4 (12.90%)		
WBC (x109)	8.02 ± 3.41	6.83 ± 1.85	0.079	
CRP (mg/L)	17.00 (7.05–47.50)	4.40 (0.65–11.10)	<0.001	
Albumin (g/L)	34.95 ± 5.76	37.04 ± 6.38	0.131	
HR (beats per minute)	89.93 ± 17.46	81.27 ± 13.88	0.022	
HRV indices				
TP (ms2)	871.72 (279.75–1595.97)	1815.11 (804.70–3397.47)	0.002	
LF (ms2)	157.41 (61.66–444.74)	378.64 (211.46–962.09)	0.006	
HF (ms2)	230.89 (36.09–450.41)	460.36 (144.39–1048.96)	0.008	
SDNN (ms)	34.55 (18.54–44.11)	48.22 (31.80–63.33)	0.005	
Note:

Abbreviations: CDAI, Crohn’s disease activity index; BMI, body mass index; WBC, white blood cell; CRP, C-reactive protein; HR, heart rate; HRV, heart rate variability; TP, total power; LF, low frequency power; HF, high frequency power; SDNN, standard deviation of R-R intervals.

Univariate analysis

The results of univariate analysis are shown in Table 3. Univariate analysis showed that BMI, HRV parameters, HR, CRP, WBC, and medications were related to the CDAI. However, gender, age, age at diagnosis, disease location, disease behavior, level of education, marital status, history of bowel resection, duration of disease, and albumin were not related to the CDAI.

Table 3 The results of univariate analysis.

CDAI	Statistics	β (95% CI)	P-value	
Age (years)	28.84 ± 11.87	0.20 [−1.68 to 2.08]	0.8325	
Gender (n, %)				
Males	60 (73.17%)	Ref		
Females	22 (26.83%)	18.60 [−31.27 to 68.47]	0.4669	
BMI (kg/m2)	18.87 ± 3.36	−13.51 [−19.44 to −7.57]	<0.0001	
Marital status (n, %)				
Unmarried	53 (64.63%)	Ref		
Married	29 (35.37%)	−14.90 [−61.16 to 31.35]	0.5296	
Level of education (n, %)				
Under middle school education	20 (24.39%)	Ref		
Senior high school education	20 (24.39%)	−19.50 [−82.98 to 43.97]	0.5487	
University degree or equivalent	42 (51.22%)	−28.17 [−82.70 to 26.37]	0.3144	
Duration of disease (years)	4.36 ± 5.11	0.88 [−3.48 to 5.24]	0.6932	
History of bowel resection (n, %)				
No	59 (71.95%)	Ref		
Yes	23 (28.05%)	0.03 [−49.32 to 49.39]	0.9989	
Medications (n, %)				
Mesalamine	45 (54.88%)	Ref		
Biologics	32 (39.02%)	−97.41 [−138.87 to −55.94]	<0.0001	
Immunomodulators	5 (6.10%)	−28.36 [−112.89 to 56.17]	0.5128	
Age at diagnosis (n, %)				
A1	14 (17.07%)	Ref		
A2	59 (71.95%)	−29.80 [−89.43 to 29.82]	0.3302	
A3	9 (10.98%)	−8.44 [−94.13 to 77.26]	0.8475	
Disease location (n, %)				
L1	19 (23.17%)	Ref		
L2	7 (8.54%)	28.30 [−58.89 to 115.50]	0.5265	
L3	53 (64.63%)	54.35 [1.62, 107.08]	0.5265	
L4	3 (3.66%)	−12.19 [−134. to 70, 110.33]	0.8459	
Disease behavior (n, %)				
B1	53 (64.63%)	Ref		
B2	13 (15.85%)	−4.26 [−66.15 to 57.62]	0.8929	
B3	16 (19.51%)	35.58 [−21.45 to 92.62]	0.2250	
WBC (x109)	7.57 ± 2.96	9.32 [2.08 to 16.56]	0.0136	
CRP (mg/L)	24.94 ± 35.22	1.06 [0.47 to 1.65]	0.0007	
Albumin (g/L)	35.74 ± 6.05	−2.68 [−6.32 to 0.96]	0.1528	
HR (beats per minute)	86.66 ± 16.66	2.50 [1.28 to 3.72]	0.0001	
HRV indices				
TP (ms2)	1,746.84 ± 1,918.12	−0.02 [−0.03 to −0.01]	0.0025	
LF (ms2)	535.83 ± 748.11	−0.03 [−0.06 to 0.00]	0.0025	
HF (ms2)	538.76 ± 785.33	−0.04 [−0.07 to −0.01]	0.0047	
SDNN (ms)	40.33 ± 21.82	−1.86 [−2.80 to −0.92]	0.0002	
Note:

Abbreviations: CDAI, Crohn’s disease activity index; BMI, body mass index; WBC, white blood cell; CRP, C-reactive protein; HR, heart rate; HRV, heart rate variability; TP, total power; LF, low frequency power; HF, high frequency power; SDNN, standard deviation of R-R intervals.

Relations between HRV indices and CDAI

The relationship between HRV parameters and CDAI was assessed using a univariate linear regression model, with both unadjusted and adjusted model results presented in Table 4. In the unadjusted model, HRV parameters including TP, HF, and SDNN were negatively correlated with CDAI (β = −0.02, 95% confidence interval [CI] [−0.03 to −0.01], P = 0.0025; β = −0.04, 95% CI [−0.07 to −0.01], P = 0.0047; β = −1.86, 95% CI [−2.80 to −0.92], P = 0.0002, respectively). In the minimally adjusted model (adjusted for age and gender), the results did not change significantly (β = −0.02, 95% CI [−0.03 to −0.01], P = 0.0029; β = −0.04, 95% CI [−0.07 to −0.01], P = 0.0065; β = −2.02, 95% CI [−3.03 to −1.02], P = 0.0002, respectively). However, in the fully adjusted model (adjusted for age, gender, medications, BMI, history of bowel resection, and disease location), the negative correlations between TP, HF, and SDNN and CDAI were not significant (β = −0.01, 95% CI [−0.02 to 0.01], P = 0.3465; β = −0.01, 95% CI [−0.04 to 0.01], P = 0.3616; β = −0.94, 95% CI [−1.88 to 0.00], P = 0.0529).

Table 4 Relationship between HRV indices and CDAI in different models.

Variable	Unadjusted mode (β, 95% CI, P)	Minimally adjusted model (β, 95% CI, P)	Fully adjusted model (β, 95% CI, P)	
TP	−0.02 [−0.03 to −0.01] 0.0025	−0.02 [−0.03 to −0.01] 0.0029	−0.01 [−0.02 to 0.01] 0.3465	
TP tertile				
Low	Ref	Ref	Ref	
Middle	−89.13 [−137.85 to −40.41] 0.0006	−94.43 [−143.85 to −45.01] 0.0003	−71.86 [−116.89 to −26.083] 0.0026	
High	−107.56 [−155.84 to −59.28] <0.0001	−116.12 [−167.26 to −64.98] <0.0001	−56.68 [−104.74 to −8.63] 0.0237	
P for trend	<0.0001	<0.0001	0.0306	
LF	−0.03 [−0.06 to 0.00] 0.0645	−0.03 [−0.06 to 0.00] 0.0837	−0.00 [−0.03 to 0.02] 0.7983	
LF tertile				
Low	Ref	Ref	Ref	
Middle	−102.39 [−151.06 to −53.71] <0.0001	−104.51 [−153.77 to −55.25] <0.0001	−69.69 [−114.39 to −25.00] 0.0032	
High	−97.76 [−146.00 to −49.52] 0.0002	−102.75 [−154.38 to −51.12] 0.0002	−48.30 [−95.82 to −0.77] 0.0503	
P for trend	0.0003	0.0002	0.0605	
HF	−0.04[−0.07 to −0.01] 0.0047	−0.04 [−0.07 to −0.01] 0.0065	−0.01 [−0.04 to 0.01] 0.3616	
HF tertile				
Low	Ref	Ref	Ref	
Middle	−67.65 [−118.52 to −16.78] 0.0109	−72.58 [−124.97 to −20.19] 0.0082	−44.53 [−89.02 to −0.05] 0.0537	
High	−90.42 [−140.84 to −40.01] 0.0007	−95.63 [−149.17 to −42.10] 0.0008	−47.68 [−94.91 to −0.45] 0.0518	
P for trend	0.0008	0.0008	0.0543	
SDNN	−1.86 [−2.80 to −0.92] 0.0002	−2.02 [−3.03 to −1.02] 0.0002	−0.62 [−1.60 to 0.37] 0.2230	
SDNN tertile				
Low	Ref	Ref	Ref	
Middle	−67.68 [−117.36 to −18.00] 0.0092	−77.18 [−128.59 to −25.77] 0.0043	−69.39 [−114.24 to −24.55] 0.0034	
High	−104.55 [−153.79 to −55.31] <0.0001	−115.74 [−168.54 to −62.94] <0.0001	−64.73 [−111.61 to −17.84] 0.0086	
P for trend	<0.0001	<0.0001	0.0101	
Note:

Unadjusted model: we did not adjust other covariants Minimally adjusted model: we adjusted age and gender Fully adjusted model: we adjusted age, gender, medications, BMI, history of bowel resection,disease location CI, confidence interval; Ref, reference.

Abbreviations: CDAI, Crohn’s disease activity index; BMI, body mass index; HRV, heart rate variability; TP, total power; LF, low frequency power; HF, high frequency power; SDNN, standard deviation of R-R intervals.

In the unadjusted model, the negative correlation between HRV parameters, LF, and CDAI was not significant (β = −0.03, 95% CI [−0.06 to −0.00], P = 0.0645). There was no change in the minimally adjusted model (β = −0.03, 95% CI [−0.06 to −0.00], P = 0.0837). In the fully adjusted model, the negative correlation between LF and CDAI was still not significant (β = −0.00, 95% CI [−0.03 to 0.02], P = 0.7983).

For the trend test, TP, LF, HF, and SDNN were used as categorical variables (tertiles). The results showed that as the value of TP and SDNN increased the value of the CDAI decreased, and the negative correlation trends were significant (P = 0.0306 and P = 0.0101).

Analyses of non-linear relations

Since HRV parameters are continuous variables, it was necessary to perform non-linear relations analyses. The results showed that after adjusting for age, gender, BMI, disease location, history of bowel resection, and medications there were non-linear relations between TP, LF, SDNN, and the CDAI, while there was a linear relation between HF and the CDAI (Fig. 2).

Figure 2 Linear or non-linear relations between HRV parameters and the CDAI after adjusting for age, gender, BMI, medications, disease location, and history of bowel resection.

A two-piecewise linear regression model was employed to determine the inflection points of the curves (as shown in Table 5). For TP, the inflection point was observed at 246.68 ms2, with an effect value of −0.77 on the left side of the point (95% CI [−1.15 to −0.4], P = 0.0001). For LF, the inflection occurred at 60.74 ms2, where the effect value on the left side was −3.95(95% CI [−5.63 to −2.27], P = < 0.0001). For SDNN, the inflection point was at 28.63 ms, and the effect value to the left of this point was −6.74 (95% CI [−9.88 to −3.59], P < 0.0001). Notably, no correlation was found between TP, LF, SDNN, and the CDAI on the right side of the respective inflection points (β = −0.00, 95% CI [−0.01 to 0.01], P = 0.9523; β = 0.01, 95% CI [−0.02 to 0.03], P = 0.5170; β = 0.46, 95% CI [−0.67 to 1.59], P = 0.4237).

Table 5 The results of two-piecewise linear regression model.

	Inflection point	Effect size (β)	95% CI	P-value	
TP	≤246.68	−0.77	[−1.15 to −0.40]	0.0001	
>246.68	−0.00	[−0.01 to 0.01]	0.9523	
LF	≤60.74	−3.95	[−5.63 to −2.27]	<0.0001	
>60.74	0.01	[−0.02 to 0.03]	0.5170	
SDNN	≤28.63	−6.74	[−9.88 to −3.59]	<0.0001	
>28.63	0.46	[−0.67 to 1.59]	0.4237	
Note:

Abbreviations: CDAI, TP: total power; LF, low frequency power; SDNN, standard deviation of R-R intervals.

Discussion

The aim of this study was to explore the relations between HRV parameters and the CDAI. Through several adjustment models and trend tests, we observed that HRV parameters were generally negatively correlated with the CDAI, among which TP, HF, and SDNN were significantly negatively correlated with the CDAI. Smooth curve fitting showed that there was a linear relation between HF and the CDAI: as HF increased, the CDAI decreased. However, there were non-linear relations between TP, LF, and SDNN and the CDAI. On the left side of the inflection point, TP, LF, and SDNN increased while the CDAI decreased, but on the right side of the inflection point there were no significant correlations.

HRV is the most sensitive index that reflects autonomic nervous function. HRV parameters are mainly divided into the time domain indics and the frequency domain indics. The time domain index SDNN refers to the standard deviation of the RR interval, which reflects the total variability of the autonomic nervous system, including the sympathetic nerves and the vagus nerve (Berntson et al., 1997; Thayer, Yamamoto & Brosschot, 2010). In the frequency domain, high frequency power HF is recognized as a vagus nerve-mediated HRV parameter (Thayer, Yamamoto & Brosschot, 2010), while low frequency power LF is considered to represent sympathetic nerve activity. However, it is involved in the baroreflex mechanism regulated by sympathetic nerves and the vagal efferent nerve, so in theory it is also affected by the vagus nerve (Goldstein et al., 2011). TP refers to the total power, which is the sum of high frequency power (HF), low frequency power (LF), and very low frequency power (VLF) (Task force of the European society of cardiology and the North American society of pacing and electrophysiology, 1996). Each HRV parameter reflected the activity of different autonomic nerves, however, it might be due to vagus nerves (Thayer, Yamamoto & Brosschot, 2010). In this study, HRV parameters were in general negatively correlated with the CDAI, reflecting that autonomic nervous function decreased due to active inflammation. Among them, TP, HF, and SDNN, which are greatly influenced by the vagus nerve, showed significant and stable negative correlations with active inflammation. However, LF parameters, which mainly represent sympathetic nerve activity, showed no significant negative correlation with the CDAI, which confirmed the physiological relation between the vagus nerve and active inflammation (Tracey, 2002).

As the main HRV parameter mediated by vagus nerve, HF exhibited the most stable and strong negative correlation with CDAI in this study, which suggests that the vagus nerve might be the common mechanism of the continuous negative correlation between HRV and inflammation. SDNN and TP also exhibited a negative correlation with active inflammation, since they contain components of the vagus nerve. A meta-analysis in 2019 showed that HRV parameters were generally negatively correlated with inflammatory markers, among which HF and SDNN had the strongest correlation with inflammatory markers (Williams et al., 2019). The results suggested that HRV parameters could be used to index activity of the neurophysiological pathway responsible for adaptively regulating inflammatory processes in humans (Williams et al., 2019). Moreover, an earlier CD study showed that there was a negative relation between HF and serum tumor necrosis factor-alpha (TNF-α) levels. Compared with CD patients with low HF, CD patients with high HF had lower TNF-α levels6. Our results are basically consistent with these results, emphasizing the importance of the vagus nerve in regulating the inflammatory response of CD.

Previous studies have shown that the decline of autonomic nervous function in CD patients mainly presents as a sympathetic-vagal imbalance (Sharma et al., 2009; Kim, Yao & Ju, 2020; Aghdasi-Bornaun et al., 2018). Curve fitting showed that TP and SDNN had a relation with the CDAI, but their relation was not significant on the right side of the inflection point. Therefore, we speculate that the main reason for the sympathetic-vagal imbalance might be a decline of vagus nerve function rather than excessive sympathetic nervous function. This physiological imbalance provides a target for CD therapy; improving vagus nerve activity may help reduce inflammatory activity.

This study is the largest sample-sized investigation to date on the relationship between HRV parameters and the inflammatory activity of Crohn’s disease. However, there are some limitations to this study. Firstly, the research was not conducted in a blinded manner; instead, the HRV parameters were collected from the system’s backend by an operator independently. Secondly, since this study was cross-sectional, it does not allow for the establishment of causal relationships.

Conclusion

This study, utilizing generalized additive models and two-segment regression analysis, uncovers the numerical relationship between heart rate variability parameters and the CDAI, emphasizing the negative linear correlation between vagal nerve activity and inflammatory activity in CD. In summary, the findings of this study reveal an association between reduced vagal nerve activity and the disease activity in patients with CD. This breakthrough provides valuable quantitative metrics for monitoring inflammatory activity and lays the theoretical groundwork for the development of vagus nerve modulation therapies.

Supplemental Information

Supplemental Information 1 Data.

Supplemental Information 2 STROBE checklist.

Additional Information and Declarations

Competing Interests

The authors declare that they have no competing interests.

Author Contributions

Beibei Wang conceived and designed the experiments, analyzed the data, authored or reviewed drafts of the article, and approved the final draft.

Shanying Liao performed the experiments, prepared figures and/or tables, and approved the final draft.

Linlin Huang performed the experiments, prepared figures and/or tables, and approved the final draft.

Zhongwen Zheng performed the experiments, prepared figures and/or tables, and approved the final draft.

Can Cui conceived and designed the experiments, analyzed the data, prepared figures and/or tables, authored or reviewed drafts of the article, and approved the final draft.

Human Ethics

The following information was supplied relating to ethical approvals (i.e., approving body and any reference numbers):

This study was approved by Research Ethics Committee of Guangdong Provincial People’s Hospital, Guangdong Academy of Medical Sciences (NO.GDREC2015048H, Date: Jan 22, 2015). Written informed consent was obtained from the patients’ legal guardians, authorizing the use of their data for research purposes.

Data Availability

The following information was supplied regarding data availability:

Raw data is available in the Supplemental Files.

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
