# Peer review of "Relationship between heart rate variability parameters and inflammatory activity in patients with Crohn’s disease: a retrospective study"

_PeerJ, doi:10.7717/peerj.19893_

## Round 0.1 · original submission · Minor Revisions

Both reviewers have provided positive comments and suggested minor revisions. Please address their comments and submit a revised version of your manuscript.

Reviewer 1 ·

Basic reporting

The paper titled “Relationship between heart rate variability parameters and inflammatory activity in patients with Crohn's disease:a retrospective study” is very interesting. Current evidence suggests that reduced vagal nerve activity in patients with Crohn's disease (CD) is related to inflammatory activity, and the HRV parameter HF, associated with the vagal nerve, shows the strongest and most stable negative correlation with inflammatory activity. However, there are a few minor issues that, if addressed, would significantly improve the manuscript.

1) The current abstract briefly reports the negative linear correlation between HF and CDAI, but it does not highlight the new findings or incremental contributions of this study compared to existing literature (e.g., Engel, Yerushalmy-Feler, etc.). It is recommended to further emphasize the unique perspectives of this study (such as the discovery of a nonlinear relationship threshold) and its clinical significance in the conclusion.

Experimental design

2)The article does not provide specific details on the diagnostic criteria for CD.

3) The article does not explain how the sample size was determined or which confounding variables were included in the study.

4) The paper includes confounding factors such as "medications," "disease location," and "history of bowel resection" in the multivariate model but does not explain why these variables were chosen or whether multicollinearity exists.

5) The baseline characteristic table lacks sufficient detail, and dropout (lost-to-follow-up) analysis is not mentioned.

Validity of the findings

What is the impact of the relationship between HF and CDAI on the treatment and prognosis of CD?

Additional comments

Some references are missing journal names, volume/issue pages, or DOIs. In addition, are there any new studies on HRV and CD since 2022 that could be included? It is recommended to update the related literature from the past two years to ensure the timeliness of the background review.

Reviewer 2 ·

Basic reporting

no comment

Experimental design

no comment

Validity of the findings

no comment

Additional comments

This study retrospectively analyzed the relationship between heart rate variability (HRV) parameters and inflammatory activity in patients with Crohn's disease (CD). The authors found that HRV parameters were generally negatively correlated with the Crohn’s disease activity index (CDAI), among which total power (TP), high-frequency power (HF), and standard deviation of R-R intervals (SDNN) were significantly negatively correlated with the CDAI. These findings provide evidence of clinical data for understanding the involvement of the cholinergic anti-inflammatory pathway of the vagus nerve in the inflammatory processes of CD. Overall, this research is a well-written paper with a clear thesis, rigorous methodology, logical structure and clear conclusion. However, some suggestions may be addressed to improve the manuscript.
1. The Result of Abstract needs to be revised, which should be descripted more detailed.
2. Abbreviations should be defined in the text at first use. If abbreviation used in Tables and Figures, the full name should be supplemented in Figure/Table legend. The author should check the whole manuscript.
3. “All patients with CD treated at our hospital from February 1, 2015 to February 1, 2020 were included.” Five years have passed. The clinical information of the patients is somewhat outdated. Why did this study take such a long time? Can the clinical data of CD patients of recent 2-3 years be supplemented?
4. The authorization information of Ethics Committee was repeated in the Materials and Methods section.
5. There are some errors concerning format. Such as, the format of parentheses should be Time New Roman (Table 2, Age and Gender). The 2 in Kg/m2 needs superscript (Table 2). There should be spaces between words and parentheses. The authors should check the full manuscript, Figures and Tables carefully.
6. In my opinion, the CDAI score in Table 2 should be moved to first line. There are 37 patients in CDAI>150 group used mesalamine and only 8 patients in CDAI≤150 group used mesalamine, which should have statistical difference. The author should add relevant description in the manuscript.
7. The word size in Figure 2 is too small, which need to be revised.
8. The overall language of the article is good. But there are still a few details that need attention. The authors should check the full text more carefully.

Annotated reviews are not available for download in order to protect the identity of reviewers who chose to remain anonymous.

---

## Round 0.2 · accepted · Accept

Both reviewers have recommended acceptance of your revised manuscript. Based on their positive feedback, I am pleased to inform you that your paper has been accepted for publication. Thank you for your careful revisions.

Reviewer 1 ·

Basic reporting

no comment

Experimental design

no comment

Validity of the findings

no comment

Additional comments

no comment

Reviewer 2 ·

Basic reporting

no comment

Experimental design

no comment

Validity of the findings

no comment

Additional comments

no comment